# Two-Period Guidance Diffusion Models for Hierarchical Conditional Generation

## Abstract

Denoising diffusion models excel at conditional generation but face a trade-off under classifier-free guidance: large guidance scales improve semantic alignment, yet reduce diversity and cause distortions, especially when there exist hierarchical structures in the conditions. We propose Two-Period Guidance Diffusion (TPGD), a simple strategy that adapts the hierarchical guidance across the denoising process. More specifically, TPGD applies coarse guidance in early steps to establish global structure, then switches to stronger guidance in later steps to refine details. Analysis under a Gaussian mixture model shows that TPGD achieves better alignment with the target distribution than standard guidance. Experiments on text-to-image benchmarks further demonstrate that TPGD consistently enhances semantic fidelity while preserving diversity, providing a principled and effective alternative to fixed-scale guidance.

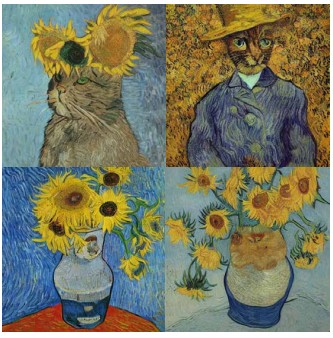 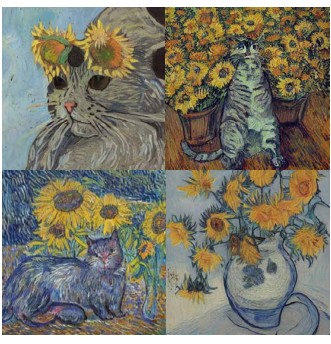 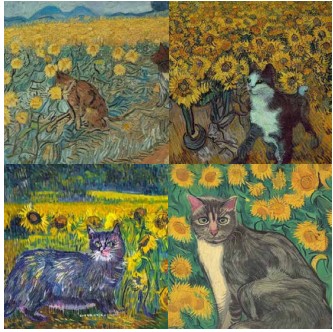

**(a)Classifier-free guidance**    **(b)Guidance interval**    **(c)Our method(TPGD)**

Figure 1: Comparison of TPGD with two baseline methods under the text prompt "A Van Gogh-style painting of a cat in a sunflower field." Our method achieves superior semantic fidelity in capturing multi-level semantics.

## 1 Introduction

Denoising diffusion models (Ho et al., 2020; Song et al., 2020; 2021) have recently become a dominant paradigm for generative modeling, showing impressive capabilities in producing high-quality samples across a wide range of modalities, including images (Dhariwal & Nichol, 2021; Rombach et al., 2022; Ramesh et al., 2022; Saharia et al., 2022), video (Ho et al., 2022b;a; Blattmann et al., 2023), 3D shapes (Poole et al., 2022; Jun & Nichol, 2023), and audio (Kong et al., 2020; Chen et al., 2021). Building on this foundation, conditional diffusion models extend flexibility by conditioning on diverse inputs, such as text prompts (Nichol et al., 2021; Ramesh et al., 2022; Saharia et al., 2022), reference images for editing and synthesis (Avrahami et al., 2022; Meng et al., 2021; Mokady et al., 2023), or structured spatial controls such as edge maps and other guidance signals (Zhang et al., 2023a; 2024). Within this line of research, classifier-guided diffusion models (Dhariwal & Nichol, 2021; Ho & Salimans, 2022) stand out for their ability to produce samples with exceptionally high fidelity, often rivaling or even surpassing the quality of other generative approaches.

Despite these advances, conditional diffusion models still face several limitations. For example, the generated content may exhibit unrealistic artifacts, semantic inconsistencies, or societal biases (Lučić et al., 2019; Bommasani et al., 2021; Weidinger et al., 2021; Luccioni et al., 2023). In particular, large guidance scales, while improving semantic alignment, often lead to distorted generations and reduced diversity (Zheng & Lan, 2023; Sadat et al., 2024; Chidambaram et al., 2024). Although various techniques have been proposed to mitigate this trade-off (Ouyang et al., 2022; Karras et al., 2024), a deeper theoretical understanding of the guidance mechanism in diffusion models is still lacking, making it an important and urgent direction for future research.

Research on the guidance term in diffusion models remains in its early stages. In text-to-image synthesis, text prompts act as guidance to generate semantically aligned images. Strong guidance improves text-image consistency but reduces output diversity. For instance, Wu et al. (2024) examined the effect of the guidance parameter $w$ within a Gaussian mixture framework, showing that increasing $w$ raises classification confidence, approaching 1 as $w \to \infty$, while simultaneously decreasing the differential entropy of the output, thereby limiting diversity. Similarly, Chidambaram et al. (2024) demonstrated that excessive guidance pushes samples toward the boundaries of the conditional distribution's support, potentially leading to distorted generations. Their findings suggest that while a high guidance scale can be advantageous, it must be carefully bounded, as even small score estimation errors can result in sampling outside the distribution's support when $w$ is too large. Kynkäänniemi et al. (2024) further observed that using a fixed guidance scale throughout the entire sampling process negatively impacts diversity and incurs unnecessary computational cost. They proposed applying guidance only during a limited middle interval of the process; however, the optimal interval varies across tasks and lacks a universal standard.

An intriguing observation is that conditions (e.g., prompts) used in guided diffusion models often exhibit hierarchical structures. As a result, these models do not generate all features simultaneously. During the early stages of the denoising process, the contours of objects or backgrounds gradually emerge. In the middle and later stages, the model refines details based on the text description. Specifically, the initial stages of diffusion models tend to generate the overall layout and color, the middle stages focus on structured appearances, and the final stages produce detailed textures (Zhang et al., 2023b). This observation supports the concept of "critical windows", where key features (e.g., object categories, colors) are determined within narrow denoising process intervals. Li & Chen (2024) formalized this phenomenon, showing that for strongly log-concave mixture distributions, these windows are bounded based on inter-group and intra-group separations.

Inspired by this observation, we propose a two-period guidance diffusion method (**TPGD**) to better balance alignment and diversity in conditional generation. Specifically, we apply a rough guidance prompt in the early stages of the denoising process to establish the layout of the image. Then, in the middle and later stages, we introduce the full guidance prompt to refine and complete the remaining details. We demonstrate that, under the Gaussian mixture model, TPGD outperforms classifier-free guidance diffusion by achieving higher alignment with the target distribution in the final sample. Through a series of experiments, we show that TPGD enhances semantic alignment while maintaining diversity. Furthermore, we provide experimental results across various guidance scales, highlighting the method's broad applicability in ensuring consistency for complex semantics.

## 2 BACKGROUND

### 2.1 DIFFUSION MODELS

Diffusion models (Song et al., 2020; Ho et al., 2020; Song et al., 2021) are a class of generative models that involve a forward process of adding noise to data and a reverse process that learns to denoise the data step-by-step to generate new samples. In the forward process, noise is added to a data point $x_0$ at time $t$ as follows:

$$x_t = \sqrt{\alpha_t} x_0 + \sqrt{1 - \alpha_t} \epsilon_t \tag{1}$$

where $\alpha_t$ is a decreasing sequence of diffusion schedule and $\epsilon_t \sim \mathcal{N}(0, I)$. A neural network $\varepsilon_\theta(x_t, t)$ is then trained to predict the noise $\epsilon_t$ that was added to $x_0$ as follows

$$\min_\theta \mathbb{E}_{x_0, \epsilon, t} ||\epsilon_t - \varepsilon_\theta(x_t, t)||_2^2, \tag{2}$$

which are then used in the reverse process for image generation.

Unlike DDPM (Ho et al., 2020), DDIM (Song et al., 2020) uses a deterministic inverse process for data generation that accelerates the sampling process, which can be described as follows

$$x_{t-1} = \sqrt{\alpha_{t-1}} \left( \frac{x_t - \sqrt{1-\alpha_t}\varepsilon_\theta(x_t, t)}{\sqrt{\alpha_t}} \right) + \sqrt{1-\alpha_{t-1}}\varepsilon_\theta(x_t, t) \tag{3}$$

Starting from $x_T \sim \mathcal{N}(0, I)$, the noise is gradually removed to generate new samples by applying eq. (3) for $T$ steps.

## 2.2 CLASSIFIER-FREE GUIDANCE

Classifier-free guidance (Ho & Salimans, 2022) is used to enhance the effect induced by the conditioned text $c$, without relying on an external classifier (Ho et al., 2020). Suppose we already have a pre-trained neural network $\varepsilon_\theta(x_t, t, c)$ for both the conditional and unconditional denoising diffusion models. Classifier-free guidance predicts noise for each step via a linear combination of conditional and unconditional predictions. Formally, let $c^\emptyset$ be the text embedding of null text, the classifier-free guidance prediction is calculated by

$$\tilde{\varepsilon}_\theta(x_t, t, c) = (1 + w) \cdot \varepsilon_\theta(x_t, t, c) - w \cdot \varepsilon_\theta(x_t, t, c^\emptyset) \tag{4}$$

where $w$ is the guidance scale parameter, By adjusting $w$, the influence of the condition $c$ can be controlled, thus enhancing the effect of conditioned text $c$ in the generated images.

## 3 PROPOSED METHOD

Classifier-free guidance diffusion struggles with prompts containing hierarchical semantics. For example, "A Van Gogh-style painting of a cat in a sunflower field" decomposes into three components: "Van Gogh-style painting", "cat", and "sunflower field". When the full guidance is applied throughout the denoising process, the model often overemphasizes style early on, causing semantic drift and neglecting elements like the cat. Prior work shows that the denoising process has stage-specific behavior: early steps define layout, middle steps refine structures, and later steps add details (Zhang et al., 2023b). If style dominates early, layout issues remain unresolved, leading to incoherent images. These observations suggest that conditional diffusion generation faces challenges when the conditions (e.g., prompts) exhibit hierarchical structures, defined as follows:

**Definition 1** (Hierarchical Conditions). *We say that a condition $c$ has a hierarchical structure if it can be decomposed into several higher-level conditions $c_1, \ldots, c_k$. That is, $c = \cap_{i=1}^k c_i$. This leads to a sequence of hierarchical conditions $c_1^H \supset c_2^H \supset \cdots \supset c_k^H$:*

$$c_1^H = c_1, \ c_2^H = c_1 \cap c_2, \ \ldots, \ c_k^H = c = \cap_{i=1}^k c_i.$$

### 3.1 TWO-PERIOD GUIDANCE DIFFUSION

To deal with the challenges associated with hierarchical conditions, we propose a simple method, which decomposes the guidance into stages aligned with the generative dynamics instead of using the full guidance along the whole generation process. In particular, suppose there exists a sequence of hierarchical conditions $c_1^H \supset \ldots \supset c_k^H$. Then, we divide the generation process into $k$ periods, $0 = T_0 < T_1 < \ldots < T_k = T$. For each $i = 1, \ldots, k$, the generation process over $[T_{i-1}, T_i]$ is governed by

$$\boldsymbol{x}'(t) = \boldsymbol{x}(t) + (w + 1)\nabla \log \pi_t(\boldsymbol{x}(t) \mid c_i^H) - w\nabla \log \pi_t(\boldsymbol{x}(t)), \tag{5}$$

which leads to a multi-period guidance procedure. In what follows, we focus on the case $k = 2$, yielding the **Two-Period Guidance Diffusion**. Our approach can be readily extended to $k > 2$.

The following section provides theoretical support for our method via Gaussian mixture models.

### 3.2 GUIDED DIFFUSION MODELS ON HIERARCHICAL GAUSSIAN MIXTURES

To simplify the analysis, consider a hierarchical distribution in which each component itself contains multiple Gaussian modes. For clarity, take the case with two top-level components, each having two

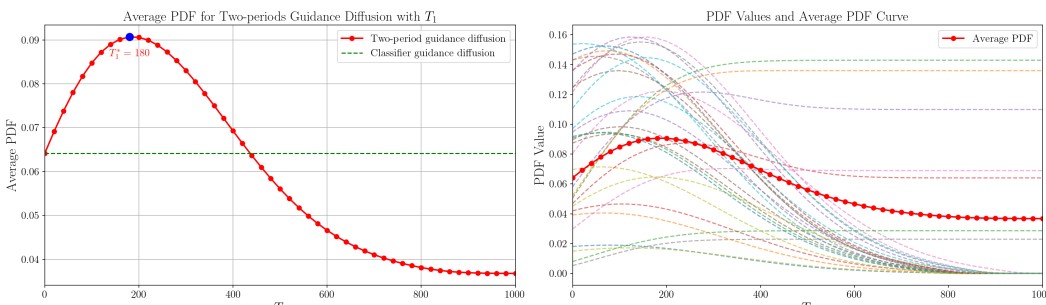

Figure 2: (Left)Grid search for $T_1$ of Type III two-period guidance diffusion. In our case, the final samples have the largest PDF when $T_1 = 180$. (Right)PDF variation with respect to $T_1$ from the particle perspective.

modes. We start from a standard Gaussian as the initial distribution and apply guidance toward one specific mode, which is treated as the target.

Traditional classifier guidance applies this attraction toward the target throughout all timesteps. This produces an "exclusion effect": trajectories are simultaneously pushed away from other modes within the same component. As a result, many final samples drift outside the actual support of the target, a behavior consistent with the findings of Chidambaram et al. (2024).

Based on the above observations, we introduce a two-period guidance strategy. In the first stage (from step 0 to $T_1$), the guidance is applied toward the broader class level, while in the second stage (from $T_1$ to step 1000), the guidance is redirected toward the specific target mode. This staged approach alleviates the exclusion effect observed with traditional classifier guidance: instead of being repelled from sibling modes, the trajectories remain within the correct class and converge more faithfully toward the target.

To evaluate the effect, we compute the average probability density of the final samples with respect to the target mode under different choices of $T_1$. As illustrated in fig. 2(a), the average density is maximized when $T_1 = 180$, indicating that this split point provides the most effective balance between class-level and target-level guidance. Notably, when $T_1 = 0$, two-period guidance reduces to classifier guidance. The average PDF of its final samples is represented by the green line in fig. 2(a). We observe that two-period guidance with $T_1 = 180$ achieves a higher average PDF than classifier guidance, indicating that our method better aligns with the support of the target distribution. We also provide, in fig. 2(b), the curves of the PDF with respect to $T_1$ for different initial samples as a reference.

### 3.2.1 THEORETICAL INSIGHT

In this section, we provide theoretical insights into the above observations, which also support the effectiveness of our approach. Specifically, we consider hierarchical Gaussian mixture models with $n = 2$ for simplicity, namely,

$$\pi_0(\boldsymbol{x}) = \frac{1}{4}\left(\mathcal{N}(\boldsymbol{x};\boldsymbol{\mu}_1, I_d) + \mathcal{N}(\boldsymbol{x};\boldsymbol{\mu}_2, I_d) + \mathcal{N}(\boldsymbol{x};\boldsymbol{\mu}_3, I_d) + \mathcal{N}(\boldsymbol{x};\boldsymbol{\mu}_4, I_d)\right).$$

Without any loss of generality, let $\{\boldsymbol{\mu}_1, \boldsymbol{\mu}_2\}$ be the first hierarchical group and $\{\boldsymbol{\mu}_3, \boldsymbol{\mu}_4\}$ be the second. In this setting, for example, sampling from the conditional distribution $\mathcal{N}(\boldsymbol{\mu}_1, \boldsymbol{I}_d)$ can be interpreted as sampling under hierarchical conditions (Definition 1):

$$\text{Sampling from } \underbrace{\text{GMM}\{\boldsymbol{\mu}_1, \boldsymbol{\mu}_2, \boldsymbol{\mu}_3, \boldsymbol{\mu}_4\}}_{\text{unconditional sampling}} \longrightarrow \underbrace{\text{GMM}\{\boldsymbol{\mu}_1, \boldsymbol{\mu}_2\}}_{\text{hierarchical condition } c_1^H} \longrightarrow \underbrace{\text{GMM}\{\boldsymbol{\mu}_1\}}_{\text{hierarchical condition } c_2^H},$$

where $\text{GMM}\{\boldsymbol{\mu}_1, \ldots, \boldsymbol{\mu}_K\}$ represents Gaussian mixture model $\pi = \frac{1}{K}\sum_{k=1}^{K}\mathcal{N}(\boldsymbol{\mu}_k, \boldsymbol{I}_d)$. To characterize the hierarchical structure of this model, we assume two properties:

**Assumption 1** (Hierarchical feature). *There exists a unit vector $\boldsymbol{v}_1$ such that*

$$\langle\boldsymbol{\mu}_1, \boldsymbol{v}_1\rangle = \langle\boldsymbol{\mu}_2, \boldsymbol{v}_1\rangle = a_1\,, \ \langle\boldsymbol{\mu}_3, \boldsymbol{v}_1\rangle = \langle\boldsymbol{\mu}_4, \boldsymbol{v}_1\rangle = a_2\,, \ a_1 \neq a_2\,.$$

**Assumption 2** (Distinguishable feature). *There exists a unit vector $\boldsymbol{v}_2^i$ such that for $i \in \{1, 2, 3, 4\}$,*

$$\langle \boldsymbol{\mu}_i, \boldsymbol{v}_2^i \rangle > \langle \boldsymbol{\mu}_j, \boldsymbol{v}_2^i \rangle, \ \forall j \in \{1, 2, 3, 4\} \backslash \{i\} \,.$$

Assumption 1 captures the similarity within each hierarchical group. The underlying idea is that two modes are projected onto the same location and the other two onto another through $\boldsymbol{v}_1$. In contrast, Assumption 2 characterizes the differences across distinct modes.

For simplicity, we consider the Ornstein–Uhlenbeck (OU) process as the forward process, which is governed by the following SDE:

$$\mathrm{d}\boldsymbol{x}_t = -\boldsymbol{x}_t \, \mathrm{d}t + \sqrt{2} \, \mathrm{d}\boldsymbol{B}_t \,, \ \text{for } t \in [0, T] \,,$$

where $\boldsymbol{x}_t \mid \boldsymbol{x}_0 \sim \mathcal{N}(e^{-t}\boldsymbol{x}_0, (1 - e^{-2t})\boldsymbol{I}_d)$. Thus, the density of the noisy distribution at time $t$ is:

$$\pi_t(\boldsymbol{x}) = \frac{1}{4} \left( \mathcal{N}(\boldsymbol{x}; e^{-t}\boldsymbol{\mu}_1, I_d) + \mathcal{N}(\boldsymbol{x}; e^{-t}\boldsymbol{\mu}_2, I_d) + \mathcal{N}(\boldsymbol{x}; e^{-t}\boldsymbol{\mu}_3, I_d) + \mathcal{N}(\boldsymbol{x}; e^{-t}\boldsymbol{\mu}_4, I_d) \right)$$

$$:= \frac{1}{4} \cdot \left( \frac{1}{\sqrt{2\pi}} \right)^d (p_1(\boldsymbol{x}, t) + p_2(\boldsymbol{x}, t) + p_3(\boldsymbol{x}, t) + p_4(\boldsymbol{x}, t))$$

Given guidance scale $w > 0$, the probability ODE of the guided diffusion is

$$\boldsymbol{x}'(t) = \boldsymbol{x}(t) + (w + 1)\nabla \log \pi_t(\boldsymbol{x}(t) \mid z) - w\nabla \log \pi_t(\boldsymbol{x}(t)) \,,$$

where $z \subseteq \{\boldsymbol{\mu}_1, \boldsymbol{\mu}_2, \boldsymbol{\mu}_3, \boldsymbol{\mu}_4\}$ is the conditional groups. Without any loss of generality, let $z$ be the first mode. Then, the guided diffusion aims to achieve sampling of $\mathcal{N}(\boldsymbol{\mu}_1, \boldsymbol{I}_d)$, i.e., the conditional distribution of $\pi_0$ conditional on the first mode. The perfect conditional sampling is characterized by the following ODE:

$$\boldsymbol{x}'(t) = \boldsymbol{x}(t) + e^{-t}\boldsymbol{\mu}_1 - \boldsymbol{x}(t) = e^{-t}\boldsymbol{\mu}_1 := \boldsymbol{g}_I(\boldsymbol{x}(t), t) \,, \ \text{for } t \in [0, T] \,, \tag{6}$$

Classifier-free guided diffusion employs a guidance scale $w > 0$. The sampling process is:

$$\boldsymbol{x}'(t) = e^{-t}\boldsymbol{\mu}_1 + w \cdot e^{-t} \cdot \left( \boldsymbol{\mu}_1 - \frac{\sum_{i=1}^{4} p_i(\boldsymbol{x}(t), t)\boldsymbol{\mu}_i}{\sum_{i=1}^{4} p_i(\boldsymbol{x}(t), t)} \right) := \boldsymbol{g}_{II}(\boldsymbol{x}(t), t) \,, \ \text{for } t \in [0, T] \,. \tag{7}$$

Our approach divides the guided generation process into two periods in order to perform conditional sampling within the hierarchical structure. In the first period, guidance corresponding to the hierarchical group, i.e., $z = \{\boldsymbol{\mu}_1, \boldsymbol{\mu}_2\}$, is used to guide the generation. In the second period, specific guidance is applied, i.e., $z = \{\boldsymbol{\mu}_1\}$. From equation 5, the overall sampling procedure is then given by

$$\begin{cases} \boldsymbol{x}'(t) = \underbrace{(w + 1) \cdot e^{-t} \cdot \frac{\sum_{i=1}^{2} p_i(\boldsymbol{x}(t), t)\boldsymbol{\mu}_i}{\sum_{i=1}^{2} p_i(\boldsymbol{x}(t), t)} \cdot e^{-t} - w \cdot e^{-t} \cdot \frac{\sum_{i=1}^{4} p_i(\boldsymbol{x}(t), t)\boldsymbol{\mu}_i}{\sum_{i=1}^{4} p_i(\boldsymbol{x}(t), t)}}_{:= \boldsymbol{g}_{III}(\boldsymbol{x}(t), t)} \,, \ t \in [0, T_1] \\ \boldsymbol{x}'(t) = e^{-t}\boldsymbol{\mu}_1 + w \cdot e^{-t} \cdot \left( \boldsymbol{\mu}_1 - \frac{\sum_{i=1}^{4} p_i(\boldsymbol{x}(t), t)\boldsymbol{\mu}_i}{\sum_{i=1}^{4} p_i(\boldsymbol{x}(t), t)} \right) \,, \ t \in [T_1, T]. \end{cases}$$
$$\tag{8}$$

We denote by $x_I(t)$, $x_{II}(t)$, and $x_{III}(t)$ the samples at time $t$ generated by the processes equation 6, equation 7, and equation 8, respectively. Then, we get the following propositions. The proof is deferred to Appendix A.

**Proposition 1.** $\langle \boldsymbol{x}_{II}(T), \boldsymbol{v}_1 \rangle = \langle \boldsymbol{x}_{III}(T), \boldsymbol{v}_1 \rangle$.

**Proposition 2.** $\langle \boldsymbol{x}_{III}(T), \boldsymbol{v}_2^1 \rangle \leqslant \langle \boldsymbol{x}_{II}(T), \boldsymbol{v}_2^1 \rangle$.

**Proposition 3.** $\langle \boldsymbol{x}_{II}(T), \boldsymbol{v}_2^1 \rangle \geqslant \langle \boldsymbol{x}_I(T), \boldsymbol{v}_2^1 \rangle$.

Proposition 1 shows that classifier-free guided diffusion and our approach achieve the same performance on generating hierarchical features. However, the next two propositions demonstrate that classifier-free guided diffusion introduces larger bias into the generation of distinguishable features, whereas our approach alleviates this issue. In particular, Proposition 2 shows that the projection of $\boldsymbol{x}_{III}(T)$ onto the distinguishable feature is smaller than that of $\boldsymbol{x}_{II}(T)$, while Proposition 3 shows that the projection of $\boldsymbol{x}_{II}(T)$ is larger than that of $\boldsymbol{x}_I(T)$, which corresponds to the correct projection of perfect samples. Since choosing $T_1 = 0$ reduces our approach to classifier-free guidance, i.e., $\boldsymbol{x}_{II}(T) = \boldsymbol{x}_{III}(T)$, setting an appropriate hyperparameter $T_1 > 0$ allows the projection of $\boldsymbol{x}_{III}(T)$ to move closer to the target $\langle \boldsymbol{x}_I(T), \boldsymbol{v}_2^1 \rangle$. Together, these three propositions demonstrate the effectiveness of our approach compared to classifier-free guidance in the hierarchical Gaussian mixture model setting.

## 4 EXPERIMENTS

### 4.1 EXPERIMENT DETAILS

All experiments were conducted on Ubuntu 22.04, using Python 3.8 and PyTorch 1.10.2 with CUDA 11.3. We used Stable Diffusion 1.5 with default hyperparameters and employed DDIM sampling with a total of $T = 1000$ steps. The denoising process was performed in 50 steps, uniformly dividing the interval from 0 to $T$ and applying denoising every 20 steps. We set the classifier-free guidance (CFG) scale to $w = 3$.

To evaluate our proposed Two-Period Guidance Diffusion (TPGD), we compared it against two text-to-image generation strategies. The first is classifier-free guidance diffusion, where the complete guidance prompt is used throughout all 1000 denoising steps. The second follows the limited-interval guidance approach proposed by Kynkäänniemi et al. (2024), where a larger guidance scale is applied within a restricted interval, while the remaining denoising steps revert to Type I conditional diffusion with $w = 0$.

### 4.2 QUALITATIVE COMPARISONS

We designed multi-level guidance prompts to examine semantic consistency in text-to-image generation. As shown in fig. 3(a), classifier-free guidance diffusion often captures only a subset of semantic information, failing to generate images with full semantic consistency.

In fig. 3(b) and (c), we compare the Guidance Interval method (Kynkäänniemi et al., 2024) and our proposed approach TPGD under the same initial noise conditions for text-to-image generation. Our method employs a two-period guidance strategy: during the first 200 denoising steps, only the basic semantic information, "A cat in a sunflower field", is provided. This allows the diffusion model to establish a coherent composition while mitigating errors introduced by the text encoder. In the remaining 800 steps, the full guidance prompt is introduced to generate specific objects, artistic style, and fine details.

The results show that, when starting from the same initial noise, our method produces images that align more closely with "a cat in a sunflower field" while simultaneously preserving the "Van Gogh-style painting" aesthetic. This significantly improves semantic alignment compared to classifier-free guidance diffusion. While the Guidance Interval method slightly enhances alignment, its synthesized images still contain incoherent elements.

**Generation Process Comparison** As shown in fig. 4, we also compared the denoising processes of the three methods starting from the same noise. It can be observed that the layout of the image is largely determined in the early stages of the denoising process; once the initial trajectory deviates, it cannot be corrected in the later stages. In contrast, fine-grained details, such as the Van Gogh style, can be rapidly injected into the generated image during the later stages of denoising. This observation directly motivates our design of Two-Period Guidance Diffusion (TPGD): the model should be guided with more accurate global semantics in the early stages to establish the overall structure, while detailed stylistic information can be incorporated in the later stages to refine the final output.

### 4.3 QUANTITATIVE COMPARISONS

To quantitatively evaluate the semantic alignment of the three methods, i.e., assessing how well the generated images adhere to the provided instructions, we consider three metrics: CLIP Score (Radford et al., 2021), ImageReward (Xu et al., 2023), and TIFA (Hu et al., 2023). CLIP Score measures the similarity between a text prompt and an image, offering a reliable estimate of how well an image corresponds to a given textual description. ImageReward assigns a preference-based score to generated images, quantifying their alignment with human judgments. TIFA generates multiple question–answer pairs for a given prompt using a large language model, and then evaluates the image by applying a visual question-answering system to answer these questions and produce a score. We report these metrics for TPGD and the other two methods in table 1.

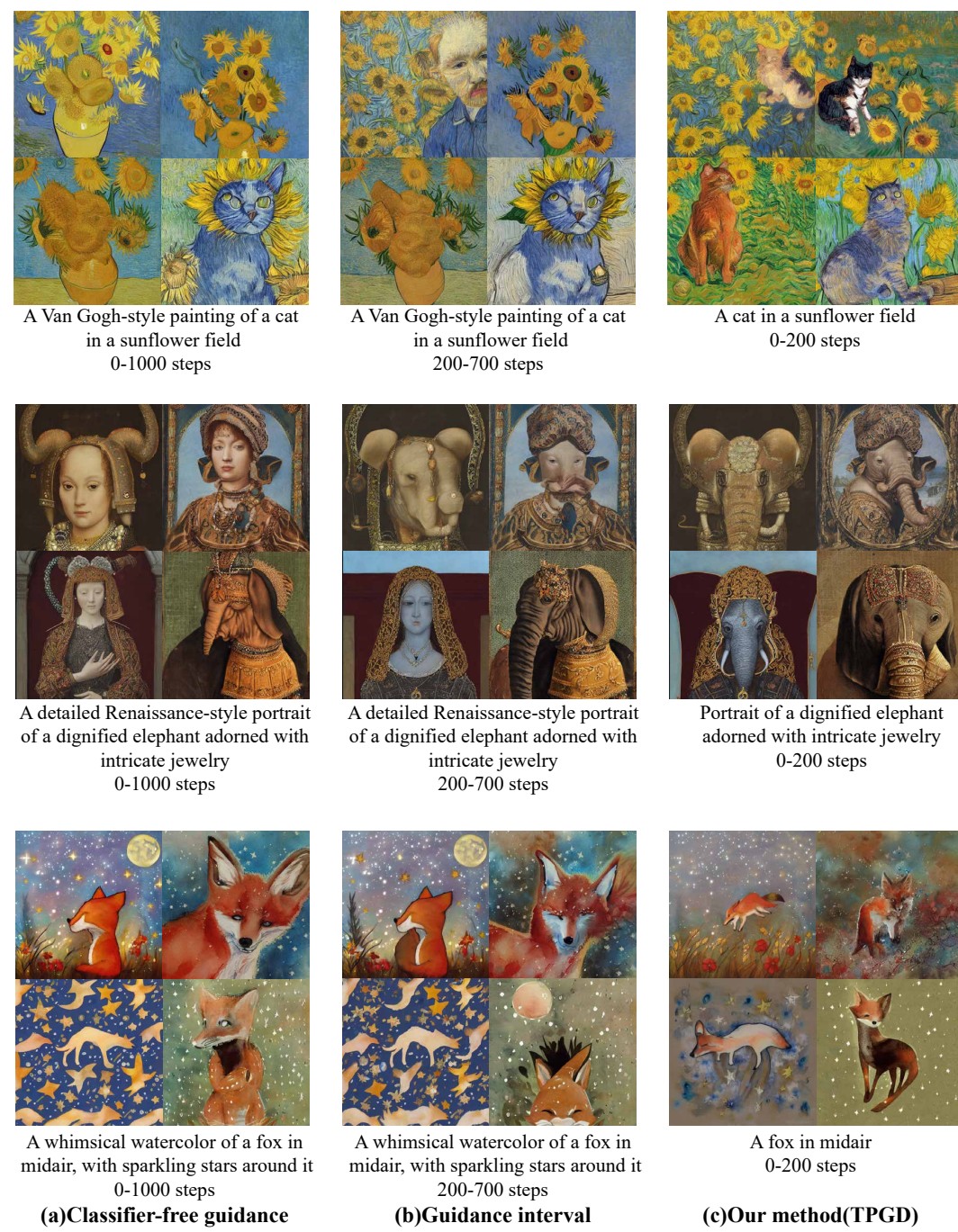

A Van Gogh-style painting of a cat in a sunflower field
0-1000 steps

A Van Gogh-style painting of a cat in a sunflower field
200-700 steps

A cat in a sunflower field
0-200 steps

A detailed Renaissance-style portrait of a dignified elephant adorned with intricate jewelry
0-1000 steps

A detailed Renaissance-style portrait of a dignified elephant adorned with intricate jewelry
200-700 steps

Portrait of a dignified elephant adorned with intricate jewelry
0-200 steps

A whimsical watercolor of a fox in midair, with sparkling stars around it
0-1000 steps
**(a)Classifier-free guidance**

A whimsical watercolor of a fox in midair, with sparkling stars around it
200-700 steps
**(b)Guidance interval**

A fox in midair
0-200 steps
**(c)Our method(TPGD)**

Figure 3: Visual comparisons of three text-to-image generation strategies. Our method achieves superior semantic fidelity in capturing multi-level semantics.

Our method outperforms the other two approaches across all three metrics. Since CLIP Score places greater emphasis on style rather than the actual content of the image, the performance differences among the three methods are relatively small under this metric. In contrast, ImageReward and TIFA are both grounded in human judgments or question–answer pairs, treating all aspects of the text prompt with equal importance. In this setting, whether the style matches is only one of many aspects being evaluated. Consequently, our method achieves a significant advantage under these

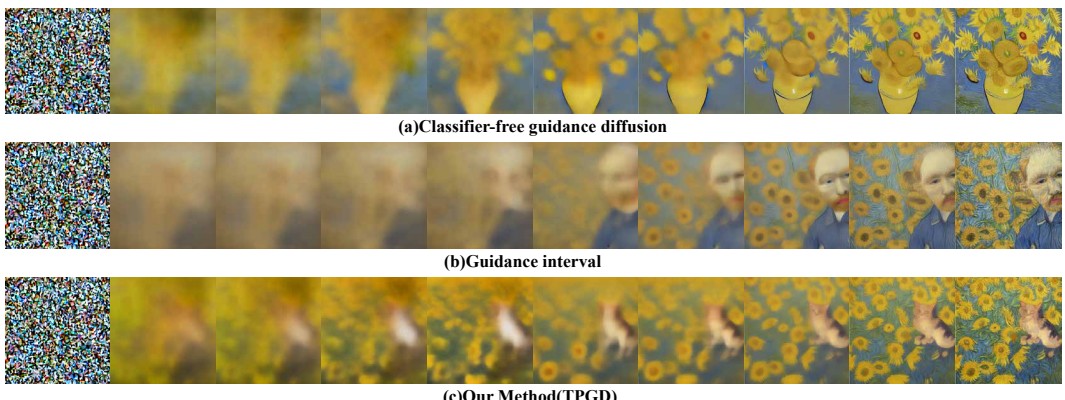

**Figure 4:** Generation process comparison. The initial layout determined in early denoising steps constrains the final image, limiting its semantic fidelity.

**Table 1:** Evaluation of semantic fidelity across multiple metrics for three text-to-image generation strategies.

| Methods | CLIP Score($\uparrow$) | ImageReward($\uparrow$) | TIFA($\uparrow$) |
|---|---|---|---|
| Classifier-free | 0.9035 | 0.1518 | 0.6500 |
| Guidance Interval | 0.9118 | 0.3483 | 0.6525 |
| Ours(TPGD) | **0.9423** | **0.9178** | **0.9100** |

two metrics, which further demonstrates that TPGD ensures faithful alignment with all aspects of the text prompt.

We further computed the average CLIP Score under different guidance scales, as shown in fig. 5. TPGD consistently outperforms the other two methods across all scales. Moreover, we report the average CLIP Score of TPGD for different timesteps $T_1$ at which the complete text prompt is introduced. The curve exhibits a trend consistent with that observed in the Gaussian mixture model experiments (fig. 2), reaching its maximum when $T_1 \in [160, 200]$. This finding indicates that, although real data are higher-dimensional and more complex, their distributions still exhibit a hierarchical structure similar to the Gaussian mixture model, thereby providing strong support for our theoretical analysis.

### 4.4 ABLATION STUDY

**Study on the impact of word order in guidance prompts.** In this section, we examine whether variations in word order within text prompts, despite conveying similar meanings, affect the text-to-image generation process in Stable Diffusion, particularly during the encoding of prompts into embeddings by the text encoder. This study is motivated by the observation that in the prompt "A Van Gogh-style painting of a cat in a sunflower field", the phrase "Van Gogh-style painting" appears at the beginning. Consequently, the text encoder may prioritize this semantic information, often leading to failures in generating the "cat".

To investigate this, we introduce an alternative prompt: "A cat in a sunflower field depicted in the style of Van Gogh" (Prompt 2), where "cat" appears earlier in the sentence structure. We compare the CLIP scores of four methods—classifier-free guidance diffusion, classifier-free guidance diffusion with Prompt 2, guidance interval, and TPGD—with respect to four different prompts. The results are presented in table 2. Consistent with our hypothesis, Stable Diffusion tends to prioritize semantic information that appears earlier in the prompt. Classifier-free guidance diffusion with Prompt 2 achieves a higher CLIP score for "A cat in a sunflower field", whereas classifier-free guidance diffusion achieves a higher CLIP score for "A Van Gogh-style painting".

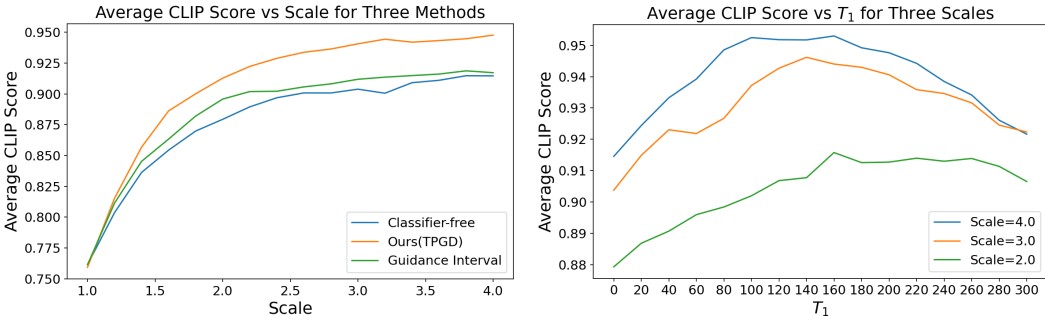

Figure 5: (Left)Average CLIP Score for different guidance scales. TPGD consistently outperforms the other two methods across all scales. (Right)Average CLIP Score of TPGD under different $T_1$ settings.

Table 2: CLIP Score results from the study on adjusting the word order in guidance prompts.

| Prompts | Classifier-free Guidance | Classifier-free Guidance with prompt 2 | Guidance Interval | Ours |
|---|---|---|---|---|
| A Van Gogh-style painting of a cat in a sunflower field | 0.9138 | 0.9370 | 0.8987 | **0.9506** |
| A cat in a sunflower field depicted in the style of Van Gogh | 0.9284 | 0.9517 | 0.9126 | **0.9660** |
| A Van Gogh-style painting | **0.7801** | 0.7685 | 0.7680 | 0.7533 |
| A cat in a sunflower field | 0.7822 | 0.8192 | 0.7678 | **0.8551** |

However, our method outperforms classifier-free guidance diffusion for both prompts in terms of CLIP scores and achieves a significantly higher CLIP score with "A cat in a sunflower field". Additionally, our method's CLIP score for "A Van Gogh-style painting" remains comparable to other methods. These findings further validate the effectiveness of our approach.

## 5 CONCLUSION

In this paper, we addressed the challenge of semantic inconsistency in classifier-free guidance when dealing with multi-level semantics in text-to-image generation. We proposed **Two-Period Guidance Diffusion (TPGD)**, which applies coarse guidance in the early denoising stages to establish the global layout and introduces full guidance in the later stages to refine semantic details. Theoretical analysis under a Gaussian mixture model demonstrates that TPGD achieves closer alignment with the target distribution compared to standard guidance. Experiments on text-to-image benchmarks further confirm that TPGD consistently improves semantic fidelity across multiple levels of prompts and guidance scales. Our work provides a heuristic approach for handling fixed text prompts, offering an effective alternative to fixed-scale guidance. Future research will focus on developing adaptive strategies for semantic layering, including how to determine which level of guidance to introduce at different stages of the denoising process, as well as conducting further theoretical analysis under alternative distance metrics.

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

**LLM Usage** In the preparation of this manuscript, LLM was used to polish grammar, style, and readability of the text.

## A    PROOF OF PROPOSITIONS IN SECTION 3

*Proof of Proposition 1.* This equality follows from for $t \in [0, T_1]$,

$$\langle g_{II}(\boldsymbol{x}(t), t), \boldsymbol{v}_1 \rangle - \langle \boldsymbol{g}_{III}(\boldsymbol{x}(t), t), \boldsymbol{v}_1 \rangle$$

$$= e^{-t} \cdot (w+1) \langle \boldsymbol{\mu}_1, \boldsymbol{v}_1 \rangle - e^{-t} \cdot (w+1) \cdot \frac{\sum_{i=1}^2 p_i(\boldsymbol{x}(t), t) \langle \boldsymbol{\mu}_i, \boldsymbol{v}_1 \rangle}{\sum_{i=1}^2 p_i(\boldsymbol{x}(t), t)} = 0 \,.$$

For $t \in [T_1, T]$, $x_{II}(t)$ and $x_{III}(t)$ share the same evolution. Hence, we conclude our proof.    □

*Proof of Proposition 2.* This inequality follows from for $t \in [0, T_1]$,

$$\langle g_{II}(\boldsymbol{x}(t), t), \boldsymbol{v}_2^1 \rangle - \langle g_{III}(\boldsymbol{x}(t), t), \boldsymbol{v}_2^1 \rangle$$

$$= e^{-t} \cdot (w+1) \cdot \left( \langle \boldsymbol{\mu}_1, \boldsymbol{v}_2^1 \rangle - \frac{\sum_{i=1}^2 p_i(\boldsymbol{x}(t), t) \langle \boldsymbol{\mu}_i, \boldsymbol{v}_2^1 \rangle}{\sum_{i=1}^2 p_i(\boldsymbol{x}(t), t)} \right) \geqslant 0 \,.$$

For $t \in [T_1, T]$, $x_{II}(t)$ and $x_{III}(t)$ share the same evolution. Hence, we conclude our proof.    □

*Proof of Proposition 3.* This inequality follows from for $t \in [0, T]$,

$$\langle g_{II}(\boldsymbol{x}(t), t), \boldsymbol{v}_2^1 \rangle - \langle g_I(\boldsymbol{x}(t), t), \boldsymbol{v}_2^1 \rangle = w \cdot e^{-t} \cdot \left( \langle \boldsymbol{\mu}_1, \boldsymbol{v}_2^1 \rangle - \frac{\sum_{i=1}^4 p_i(\boldsymbol{x}(t), t) \langle \boldsymbol{\mu}_i, \boldsymbol{v}_2^1 \rangle}{\sum_{i=1}^4 p_i(\boldsymbol{x}(t), t)} \right) \geqslant 0 \,.$$

Hence, we conclude our proof.    □

## B    VISUALIZATION ON HIERARCHICAL GAUSSIAN MIXTURES

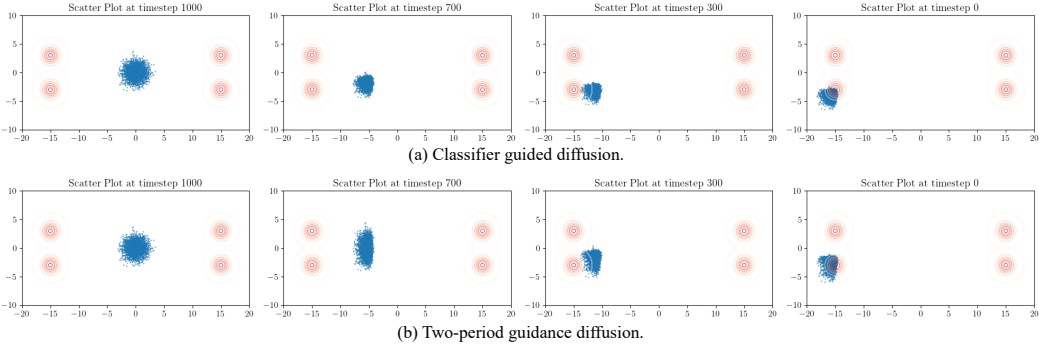

(a) Classifier guided diffusion.

(b) Two-period guidance diffusion.

Figure 6: (a)Type II classifier guidance diffusion on Gaussian Mixture Models. (b)Type III two-period guidance diffusion on Gaussian Mixture Models.

## C  VISUAL COMPARISONS OF IMAGES GENERATED WITH DIFFERENT $T_1$ VALUES.

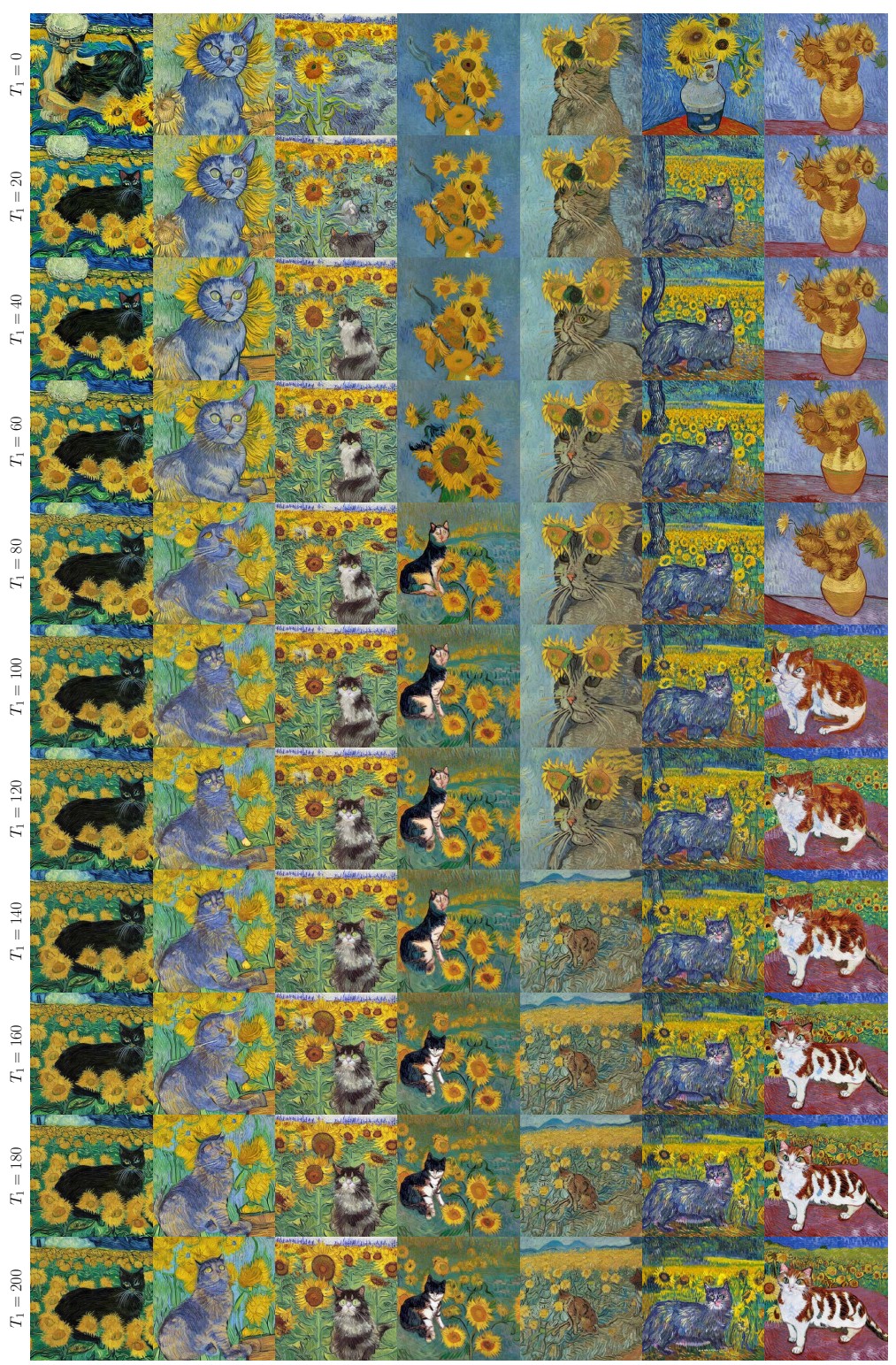

Figure 7: Visual comparisons of images generated by our proposed TPGD with different $T_1$ values.

