# OpenReview forum: "Two-Period Guidance Diffusion Models for Hierarchical Conditional Generation"
_ICLR.cc/2026/Conference — Submitted to ICLR 2026_

### Official Review · Reviewer_hQ1Z · 2025-10-20

**Soundness:** 3
**Presentation:** 3
**Contribution:** 3
**Rating:** 2
**Confidence:** 5

**Summary:**

The paper proposes to separate the whole guided sampling process into several steps with hierarchical conditions, which mainly bases on the conclusion that denoising process has stage-specific behavior. By doing so, the model could avoid to overemphasize the style component in the text prompt and synthesize samples with better fidelity. Both theoretical and experimental analyses confirm the proposed method.

**Strengths:**

- The idea to separate the denoising process with hierarchical conditions is novel and neat.
- The proposed method is easy to implement.
- The experiments are extensive, confirming the superiority of the proposed method.

**Weaknesses:**

- The paper needs further proofreading. The current version is poorly written and hard to follow.
  1. The authors use "classifier guidance" at the beginning of Sec. 3.2 and "Classifier-free" in Tab. 1, which should be "classifier-free guidance".
  2. By Def. 1 in L144, there is no unconditional stage, which conflicts with L210.
  3. $T$ means the ending timestep of diffusion process in both Sec. 2.1 and L226 which conflicts with L151, so it should be $x_{II}(0)$ and $x_{III}(0)$ in L257-259, and $T_1=T$ reduces to native CFG. This mistake makes paper hard to understand.

- The proposed pipeline is not sufficiently studied. The authors claim that the denoising process could be separated into multi-period procedure but choose to use only two steps. This is somewhat intuitive since the authors mainly focus on "style" and "object" components. However, the functionality of this methodology is not well explored. Besides, the ablation on $T_1$ is poor and not convincing.
  1. Effect of different way to separate the conditions.
  2. What if no "style" component or several "style"?
  3. How about separating "object" components into a hierarchical manner?

- The theoretical analysis is not convincing. The inequalities in Prop. 2 and 3 are not compact enough, *i.e.*, to what extent the bias is reduced? Besides, the visualization in Fig. 6 also demonstrates similar behaviors and no significant improvements.

- The ablation of word order seems irrelevant with the whole paper. Besides, Prompt 2 refers to as the alternative prompt in a different order, then what is the "prompt 2" of line 2 in Tab. 2? From my perspective, prompt in line 2 is the "prompt 2" of that in line 1.

**Questions:**

See Weaknesses part.

---

### Official Review · Reviewer_XGHY · 2025-10-29

**Soundness:** 2
**Presentation:** 3
**Contribution:** 1
**Rating:** 2
**Confidence:** 4

**Summary:**

This paper aims to solve a problem in text-to-image generation: when using classifier-free guidance, complex and hierarchical text prompts often lead to semantic inconsistencies.

To tackle this issue, the authors introduce the TPGD method, employing coarse-grained conditions in the initial phases and fine-grained conditions in the subsequent phases. They assert that this strategy mitigates conflicts between competing concepts early in the generation process, with theoretical validation provided through a simple Gaussian mixture model analysis.

**Strengths:**

1. The writing is clear and well-structured, with a detailed explanation of the methodological process.

2. The inclusion of numerous visualization cases makes the content highly accessible and reader-friendly.

**Weaknesses:**

1. The biggest issue with this paper is the absence of a **related work section**, which means the authors fail to provide comparisons and discussions with prior methods. In fact, the concept of diffusion models generating layouts and contours in early stages and refining details in later stages is not novel [1]. Moreover, switching guidance (e.g., prompts) at different stages of the generation process is not a new approach either (Especially in the field of image editing.). The lack of novelty makes it difficult to convincingly support their claims.

2. The authors' theoretical analysis is built upon a Gaussian mixture model with **very strong assumptions** (featuring four Gaussian sub-distributions arranged in a progressively hierarchical structure). However, there exists a significant gap between this toy scenario and real-world text-to-image tasks. If the authors believe such a simplified Gaussian mixture can adequately represent text-to-image scenarios, **I would recommend they further develop this simplified model by incorporating analyses of upper or lower bounds**—for instance, an analysis of the lower bound—to demonstrate the specific quantitative improvements achieved by TPGD.

3.The TPGD only compares with interval guidance, and the baseline model is limited to SD 1.5. Such a limited experimental setup is insufficient to be convincing. Since SD 1.5 is a relatively early model, I recommend adopting DiT architectures for validation, such as Flux or SD 3.5.

[1] Wang, Binxu, and John J. Vastola. "Diffusion models generate images like painters: an analytical theory of outline first, details later." arXiv preprint arXiv:2303.02490 (2023).

**Questions:**

Please refer to the weaknesses section.

---

### Official Review · Reviewer_e3no · 2025-10-31

**Soundness:** 2
**Presentation:** 2
**Contribution:** 2
**Rating:** 2
**Confidence:** 5

**Summary:**

The paper proposes Two-Period Guidance Diffusion (TPGD) for conditional diffusion models with hierarchical prompts. Instead of applying full classifier-free guidance (CFG) throughout the entire denoising trajectory, TPGD uses coarse, high-level guidance in the early steps to establish global structure and switches to the full prompt in later steps to inject style and fine details. The authors analyze the method under a hierarchical Gaussian mixture model (GMM), showing that TPGD mitigates the “exclusion effect” of standard CFG and better aligns samples with the target distribution.

**Strengths:**

1. The central idea is conceptually simple and easy to grasp.
2. The paper is clearly written and well organized.

**Weaknesses:**

1.  TPGD appears tailored to text-conditional image generation where the condition can be semantically decomposed. For class-conditional image generation (and other non-text conditions), the condition is not naturally decomposable into hierarchical sub-conditions, making TPGD inapplicable or ill-defined.
2. It is unclear how the hierarchical prompts are obtained. If my understanding is correct, the semantic layers are manually crafted, which introduces potential unfairness in the comparisons. More importantly, the work feels incomplete without a clear, reproducible pipeline (or at least guidelines) for prompt decomposition.
3. Finally, the experiments are not fully convincing. For Table 1, on which benchmark are the results reported? Were the prompts re-designed for all samples in the benchmark?

**Questions:**

See Weakness

---

### Official Review · Reviewer_y1Cr · 2025-11-09

**Soundness:** 3
**Presentation:** 2
**Contribution:** 2
**Rating:** 2
**Confidence:** 3

**Summary:**

This paper extends the idea of Guidance Interval by introducing a two-period hierarchical guidance strategy. The authors first decompose the conditioning into hierarchical components and then combine them to represent the overall condition. Guidance is applied in two stages: during the early timesteps, coarse-level guidance is provided, followed by fine-grained guidance at later timesteps. Unlike previous approaches that apply a uniform guidance scale, this staged design achieves better text-image consistency and perceptual quality.

**Strengths:**

The proposed method shows a clear improvement in ImageReward scores compared to prior approaches. Qualitative results also demonstrate that for prompts combining multiple concepts, the hierarchical approach effectively captures all sub-concepts, indicating better compositional understanding. This suggests that the coarse-to-fine strategy is beneficial in guiding diffusion trajectories toward semantically coherent outcomes.

**Weaknesses:**

This paper does not sufficiently discuss prior work that already varies guidance or ControlNet scales across timesteps, as implemented in practical frameworks such as ComfyUI or ControlNet. These techniques have been used in practice to improve quality, and a deeper comparison or distinction would strengthen the contribution.

A more fundamental concern lies in how the conditional information is hierarchically decomposed. The process of splitting conditions into high-level and low-level concepts is unclear and potentially ambiguous. Defining such hierarchies often depends on constructing explicit ontologies or concept graphs, which remains a non-trivial and only partially solved problem despite recent progress in large language models. The paper should clarify how these hierarchical levels are defined or derived automatically.

Furthermore, the method seems applicable only to text-conditioned image generation, where linguistic hierarchy exists. For class-conditional generation, such as ImageNet models, this assumption breaks down: while a class like golden retriever may exist, its parent concept dog may not be explicitly represented, making the hierarchical conditioning inapplicable. Thus, the proposed approach may not generalize to broader conditional generation settings.

To claim hierarchical guidance as a main contribution, the paper should provide a more detailed discussion on prompt decomposition, how to split a prompt into hierarchical stages, how this process can be automated, and how it scales to diverse prompts without manual design.

Finally, the experimental details are insufficient. Only three prompt examples are provided, which is too limited to support the general claims. The paper should include more diverse qualitative cases and quantitative metrics such as FID to assess image quality improvements. Moreover, details such as the number of samples used for computing CLIP Score or ImageReward are missing, which affects the reliability of the reported results.

**Questions:**

Please refer to the weaknesses

---

### Meta-Review · Area_Chair_JQZK · 2025-12-21

**Summary:**

This paper proposes TPGD, a method to improve semantic alignment in text-to-image generation. The core idea is to address the trade-off in classifier-free guidance by splitting the denoising process into two stages: and early stage using “coarse” guidance and a later stage using full guidance. The authors provide a theoretical analysis based on a hierarchical Gaussian Mixture Model and experimental results on Stable Diffusion 1.5.

All reviewers are all clear in their rejection with a highly consistent score 2, and the fundamental design flaws identified in the initial reviews, specifically regarding the manual nature of the method, remain inherent toe the proposed approach and were not resolved by the revisions.

The authors gave up rebuttal.

Thus, I recommend this submission for Reject.

**Reviewer Concerns:**

All concerns are outstanding, as the authors did not participate in the rebuttal or discussion.

(1) Methodological Flaw: This is the most critical concern shared by Reviewers y1Cr and e3no. The method relies on decomposing a prompt into “hierarchical components”. The reviewers rightly pointed out that this process is undefined, potentially ambiguous, and currently manual. This paper lacks an automated pipeline to determine how to split a prompt or determine the hierarchy for arbitrary inputs, making the method impractical for general t2i generation.

(2) Limited Novelty and Scope: Reviewer XGHY noted that the insight that diffusion models generate layout early and details later is well established and not a novel contribution of this paper. Reviewer y1Cr pointed out that dynamic guidance is already used in practice, and the paper failed to discuss or compare against these existing practical. Furthermore, the method is strictly limited to text-conditional tasks where semantic decomposition is possible, falling to generalize to class-conditional generation(Reviewer e3no).

(3) Insufficient Evaluation and Theoretical Weaknesses: The experiments are conducted solely on Stable Diffusion 1.5, which is an outdated architecture. Reviewer XGHY and hQ1Z suggested that validation on modern DiT architectures is necessary. Additionally, the qualitative evaluation was deemed sparse, and quantitative details are missing. Besides, Reviewers XGHY and hQ1Z point out that the theoretical analysis based on Gaussian Mixture Models to rely on overly strong assumptions that do not bridge the gap to real-world text-to-image scenarios.

**Reviewer Scores:**

As there is no response from the authors, I believe all reviewers will maintain their rating as 2.

Reviewer y1Cr: 2 -> 2

Reviewer e3no: 2 -> 2

Reviewer XGHY: 2 -> 2

Reviewer hQ1Z: 2 -> 2

---

### Decision · Program_Chairs · 2026-01-26

Reject